# 1,8 Cineole and Ellagic acid inhibit hepatocarcinogenesis *via* upregulation of MiR-122 and suppression of TGF-β1, FSCN1, Vimentin, VEGF, and MMP-9

Heba M. I. Abdallah[1], Sally A. El Awdan[1], Rehab F. Abdel-Rahman[1], Abdel Razik H. Farrag[2], Rasha M. Allam[1]*

**1** Department of Pharmacology, National Research Centre, Dokki, Cairo, Egypt, **2** Department of Pathology, National Research Centre, Dokki, Cairo, Egypt

\* rasha_senior@yahoo.com

**Data Availability Statement:** All relevant data are within the manuscript and its Supporting Information files.

## Abstract

Hepatocellular carcinoma (HCC) is one of the most burdened tumors worldwide, with a complex and multifactorial pathogenesis. Current treatment approaches involve different molecular targets. Phytochemicals have shown considerable promise in the prevention and treatment of HCC. We investigated the efficacy of two natural components, 1,8 cineole (Cin) and ellagic acid (EA), against diethylnitrosamine/2-acetylaminofluorene (DEN/2-AAF) induced HCC in rats. DEN/2-AAF showed deterioration of hepatic cells with an impaired functional capacity of the liver. In addition, the levels of tumor markers including alpha-fetoprotein, arginase-1, alpha-L-fucosidase, and ferritin were significantly increased, whereas the hepatic miR-122 level was significantly decreased in induced-HCC rats. Interestingly, treatment with Cin (100mg/kg) and EA (60mg/kg) powerfully restored these biochemical alterations. Moreover, Cin and EA treatment exhibited significant downregulation in transforming growth factor beta-1 (TGF-β1), Fascin-1 (FSCN1), vascular endothelial growth factor (VEGF), matrix metalloproteinase-9 (MMP-9), and epithelial-mesenchymal transition (EMT) key marker, vimentin, along with a restoration of histopathological findings compared to HCC group. Such effects were comparable to Doxorubicin (DOX) (2mg/kg); however, a little additive effect was evident through combining these phytochemicals with DOX. Altogether, this study highlighted 1,8 cineole and ellagic acid for the first time as promising phytochemicals for the treatment of hepatocarcinogenesis via regulating multiple targets.

## Introduction

Hepatocellular carcinoma (HCC) is the fourth main cause of cancer-related mortality, creating a large global cancer burden [1, 2]. Unfortunately, HCC is characterized by rapid growth, high invasive potential, and early metastasis [3, 4]. Specifically, epithelial-mesenchymal transition (EMT) is partially considered as a critical step for HCC progression via mediating early

**Funding:** yes, This work was supported by a grant from the National Research Centre (NRC), Egypt (No. 11010330).

**Competing interests:** The authors have declared that no competing interests exist.

metastasis, migration, and invasion of tumor cells and acts as a potential therapeutic target in HCC treatment [5]. EMT is the process of cell remodeling that is characterized by upregulated expression of mesenchymal markers like vimentin, being one of its most important molecular features [6]. Inducers of EMT include several growth factors like vascular endothelial growth factor (VEGF) [7] and transforming growth factor beta-1 (TGF-β1) [8]. TGF-β1 is one of the pivotal factors regulating EMT, responsible for its initiation and maintenance in HCC [9]. TGF-β1 initiation is further activating multiple cellular responses. Fascin-1 (FSCN1), an actin-binding protein involved in the invasion and migration of a variety of tumors, was verified as a direct target of TGF-β1 activation in HCC, and its overexpression was correlated with vimentin upregulation [10]. In another response, several matrix metalloproteinases (MMPs) are overexpressed during EMT and because of TGF-β1 activation [11]. From all MMPs, MMP-9 showed elevated expression and is considered an essential factor in HCC being a promoter of tumor metastasis and angiogenesis as well [12]. Thanks to MMP-9 proteolytic activity, it promotes extracellular matrix (ECM) stored growth factors mobilization, including VEGF, thus favoring HCC angiogenesis. VEGF, another EMT inducer, is the most potent known angiogenic factor, and the treatment strategies of HCC targeting VEGF have become a hotspot [13].

Ellagic acid (EA) is a natural phenolic constituent present in berries, walnuts, grapes, pomegranates, black currents, and dried fruits [14]. EA is readily absorbed through the GIT to act on sub-cellular components and activate signaling transduction in target tissues and cells [15]. EA is considered as one of the most promising and applied chemopreventive agent against several tumors without causing toxicity to normal cells [16]. However, little is known about its molecular mechanisms in HCC [17]. 1,8-Cineole (also known as eucalyptol), a monoterpene found naturally in essential oils of several plants, including eucalyptus, rosemary, cardamom, and sweet basil. It has been widely reported for its anti-inflammatory, antimicrobial, antiseptic, and antioxidant bioactivities [18–20] and, to a much lesser extent, as an anticancer compound with poorly understood underlying mechanisms [21, 22].

We aimed to assess the potential anticancer effect of EA and Cin alone and when combined with the standard chemotherapeutic agent, Doxorubicin, in an in-vivo model of rat HCC. Furthermore, we initially shed light on their possible effects on EMT and its regulators.

## Materials and methods

### Drugs and chemicals

Diethylnitrosamine (DEN) (#bN0756), ellagic acid (#bE2250), doxorubicin hydrochloride (#bD1515), 1,8 cineole (#bC80601), acetylaminofluorene (2-AAF) (#bA7015) were purchased from Sigma-Aldrich Co.

### Animals and experimental design

All procedures in this work were conducted consistently with the ethical standards and protocols of the Guide for the Care and Use of Laboratory Animals and with the Animal Experimentation Ethical Committee of NRC under approval no. of 16/383. The animals were acclimatized for one week under laboratory conditions, and then rats were randomly divided into seven groups (n = 10). **Group 1 (Normal)** rats were fed with a standard diet, i.p. injected once weekly with saline, and served as a negative control. **Group 2 (Induced-HCC)** rats with DEN/2-AAF as described below and served as untreated HCC animals. **Groups 3–5 (EA, Cin, and Dox)** were DEN/2-AAF induced HCC rats treated with daily administration of ellagic acid (60 mg/kg p.o.) [23] or daily administration of 1,8 cineole (100 mg/kg p.o.) [24] or with a weekly injection of doxorubicin hydrochloride (2 mg/kg i.p.) [25], respectively for the last four weeks after HCC induction. **Group 6 and 7 (EA/Dox and Cin/Dox)** were DEN/2-AAF

induced HCC rats administered in combination with the same doses as each alone. The general health conditions of animals concerning food consumption, water intake, and body weight were observed during all the experimental time besides signs of distress including respiratory changes, body temperature, or even unusual behavior were also observed. We did not notice a significant variation in the previous parameters between the experimental groups except for the DEN/2-AAF group that showed a modest weight loss compared with the others.

## Induction of hepatocarcinogenesis

HCC was experimentally induced in rats by a single i.p. injection of 200 mg/kg diethylnitrosamine (DEN) as the initiator. After eight weeks, liver cancer development was promoted with a once-weekly administration of 2-acetylaminofluorene (2-AAF) 30 mg/kg p.o. for another three weeks according to the protocol previously described [26] with few modifications. Treatment was started in the last 4 weeks of the experiment. Throughout the initiation phase of DEN treatment (first eight weeks), only one rat was dead. After starting 2-AAF administration (promotion phase), another two rats from the DEN/2-AAF induced HCC group died.

## Sample preparations

At the end of the experiment, rats were lightly anesthetized, and blood samples were collected from the orbital sinuses and centrifuged to obtain sera for biochemical analysis. Then rats were killed by cervical dislocation under light ethyl ether anesthesia. The liver tissues were quickly taken and divided for further analyses. Serum and liver tissues were stored at −80°C until they were analyzed.

## Determination of serum liver biomarkers

The enzyme activities of alanine transaminase (ALT), aspartate transaminase (AST), and alkaline phosphatase (ALP) as well as albumin and total protein concentrations were determined in the serum by commercially available reagent kits (Biodiagnostic Co., Cairo, Egypt). The assays were done according to the supplied protocols.

## Determining HCC tumor markers

**Alpha-fetoprotein (AFP), Alpha-L-Fucosidase (AFU), Arginase-1(Arg-1), and ferritin** serum levels were determined using ELISA kits obtained from Glory Science Co., Ltd., USA. The assays were performed as mentioned by the supplied protocols.

**MiR-122 expression using quantitative RT-PCR technique.** Total RNA was extracted from liver tissue with the RNeasy kit; Qiagen, Valencia, CA, USA as prescribed by the manufacturer. Isolated RNA was instantly archived into a cDNA library via the high-capacity cDNA reverse transcription kit, Applied Biosystems, Foster City, CA, USA. Real-time PCR was performed using the SYBR Green mix (BioRad) and the Rotor-Gene 1.7.87 RT- PCR system consistent with the manufacturer's protocol. Relative gene expression was normalized to U6 small nuclear RNA and **β actin,** respectively, and was calculated using the $2^{-\Delta\Delta CT}$ method. *MiR-122* primer sequence was **F:** 5′-GGCTGTGGAGTGTGACAATG -3′ and **R:** 5′-GAGGTA-TTCGC ACCAGAGGA -3′ [27].

## Western blot analysis of vimentin, TGF-β1, FSCN1, and MMP-9

For the analysis of protein expression, liver homogenates were extracted for their protein contents by ice-cold RIPA buffer supplemented with phosphatase and protease inhibitors followed by centrifugation at 13,000 ×g for twenty min. The protein concentrations were quantified by

Bradford assay. Equal protein amounts (50 μg) were loaded in 10% SDS-PAGE and transferred to an activated PVDF membrane after electrophoresis. The membranes were blocked with 5% bovine serum albumin (BSA) solution for overnight at 4˚C. Immunoblotting was performed using monoclonal antibodies to vimentin, TGF-β1, FSCN1, and MMP-9 as described previously [28]. Afterward, the incubation with the primary antibody for three h followed by the incubation with the secondary antibody (labeled with alkaline phosphatase) for two h was done at 25˚C. Bands were visualized via the NBT-BCIP solution. The relative protein levels were determined after normalization with β-actin protein and analyzed by GS-700 imaging densitometer usingV3 Western Workflow™ Complete System software; version 1.5, Bio-Rad Laboratories, Hercules, CA, USA.

## Determining VEGF2

Its level in liver homogenates was determined using ELISA kits, Glory Science Co.

## Histopathological evaluation

The liver tissues were separated instantly, washed with ice-cold saline, and fixed in 10% formalin. After fixation, tissues were dehydrated in ascending grade of ethanol and embedded in Paraffin wax followed by sectioning at 5 μm to be mounted on slides. Then slides were oven-dried, deparaffinized, and stained with H & E staining to be viewed using a light microscope.

## Statistical analysis

The statistical analysis of data was performed with one-way ANOVA and Tukey's post-hoc test using Graph pad prism software version 6. Data are presented as mean ± SEM, n = 10. The value of ($P < 0.05$) was considered as significant. '**a**' vs. the normal group, '**b**' vs. DEN/2-AAF-induced HCC group, and '**c**' vs. DOX treated group.

## Results

### 1,8 cineole and ellagic acid protection against DEN/2-AAF induced liver damage

The induction of HCC with DEN/2-AAF resulted in a dramatic change in the functional activity of the liver represented by a significant ($P<0.05$) increase in the serum activities of ALT, AST, and ALP that was accompanied by a significant decrease in the serum concentrations of total protein and albumin when compared to normal rats as displayed in (Table 1(. Treatment with 1,8 cineole (100 mg/kg) and ellagic acid (60 mg/kg) for four weeks showed a significant ($P<0.05$) decrease in the serum activities of ALT, AST, and ALP compared with HCC untreated group. The ALP activity was noticeably decreased in the DOX-treated group compared with HCC untreated group.

The concentration levels of total proteins and albumin were significantly ($P<0.05$) increased in treated HCC rats with 1,8 cineole and ellagic acid compared with untreated HCC rats. Notably, the combination treatment of 1,8 cineole and ellagic acid with Doxorubicin did not show significant improvement in the levels of liver biomarkers when compared to the DOX-treated group.

### Effect on serum HCC tumor markers

**Alpha-fetoprotein (AFP)** is the main golden marker for HCC detection. Rats administered DEN/2-AAF showed significant ($P<0.05$) elevation in AFP level to 716.6 ± 60.5 ng/ml as compared to the control group (270 ± 10 ng/mL). All treatment groups almost normalized the AFP

**Table 1. Effect of 1,8 cineole (Cin), ellagic acid (EA), doxorubicin (DOX) and combinations on serum liver markers of DEN/2-AAF-administered rats.**

| Groups | ALT (U/L) | AST (U/L) | ALP (U/L) | Total protein (g/L) | Albumin (g/L) |
|---|---|---|---|---|---|
| Normal | 21±0.16 | 66.05±3 | 180.00±8.2 | 8.83±0.28 | 5.05±0.1 |
| HCC | 44.1±1.6 [a] | 135.7±1.7 [a] | 567.50±8.5 [a] | 5.68±0.27 [a] | 2.88±0.26 [a] |
| HCC+DOX | 35.57±0.39 [a] | 132.59±4.2 [a] | 355.60±9 [a,b] | 4.89±0.16 [a] | 2.07±0.3 [a] |
| HCC+EA | 30.05±0.7 [b] | 107.5±2.7 [b] | 401.40±5.3 [b] | 7.41±0.3 [b] | 3.71±0.16 [b] |
| HCC+EA+DOX | 31.9±0.63 [b] | 96.22±7.6 [b] | 380.20±5.1 [b] | 6.66±0.32 | 4.22±0.21 [b] |
| HCC+ Cin | 30.17±0.1 [b] | 111.3±4.1 [b] | 381.25±4.7 [b] | 7.39±0.17 [b] | 3.68±0.13 [b] |
| HCC+Cin+DOX | 28.67±1.8 [b] | 92.59±5.6 [b] | 258.00±9.4 [b] | 7.08±0.29 | 3.81±0.24 [b] |

Data are presented as mean ± S.E.M, n = 10.

[a] $P<0.05$ vs. the normal group and

[b] $P<0.05$ vs. DEN/2-AAF-induced HCC group using one-way ANOVA followed by Tukey's post-hoc test.

ALT (Alanine transaminase); AST (Aspartate transaminase); ALP (Alkaline phosphatase).

level. The level of AFP was decreased to 243.3 ± 20.1 ng/mL, 286.6 ± 21.8 ng/mL and 266.6 ± 8 ng/mL in rats treated with DOX (2 mg/kg), Cin (100 mg/kg) and EA (60 mg/kg), respectively. No significant difference in AFP levels was observed between groups compared to the Dox group, including combination groups (Fig 1A).

**Serum ferritin**, another biochemical marker of HCC and the major cellular storage protein for iron, was elevated to about two folds (234 ±4 ng/ml) after DEN/2-AAF induction of HCC compared to normal rats (112 ± 3.5 ng/mL). This elevation was opposed by all treatment

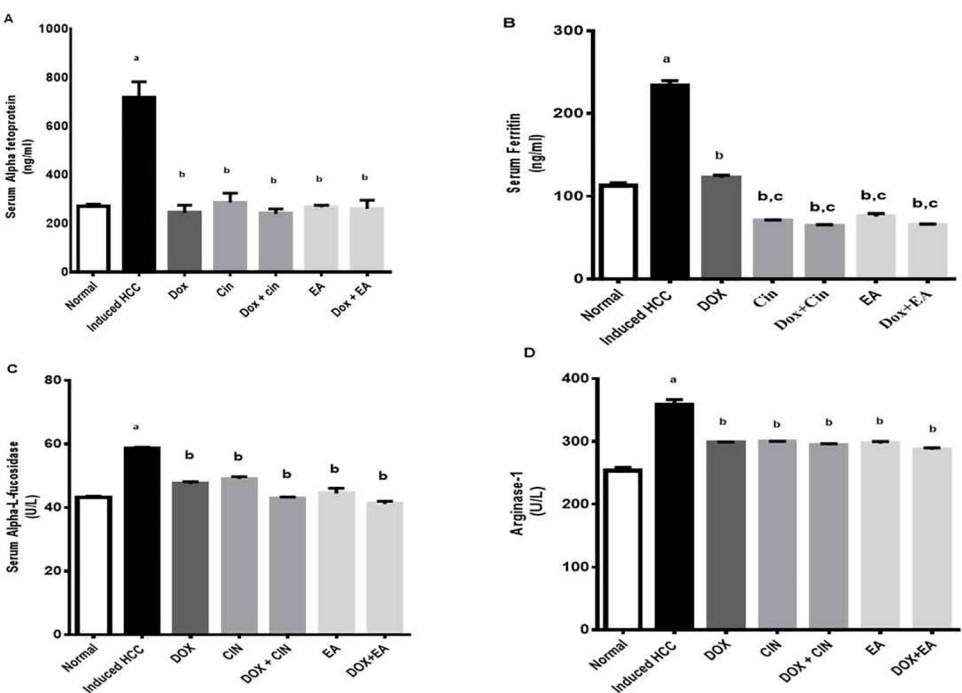

**Fig 1. Effect of 1,8 cineole (Cin), ellagic acid (EA), doxorubicin (DOX), and combinations on HCC tumor markers.** Serum levels of alpha-fetoprotein (A), ferritin (B), Alpha-L-fucosidase (C) and Arginase-1(D) were measured in DEN/2-AAF-induced HCC rats using ELISA kits. Data are presented as mean ± S.E.M, n = 10. [a] $P<0.05$ vs. normal group, [b] $P<0.05$ vs. DEN/2-AAF-induced HCC group and [c] $P<0.05$ vs. DOX treated group using one-way ANOVA followed by Tukey's post-hoc test.

groups (Fig 1B). Interestingly, serum ferritin concentrations in Cin (70.5±0.71 ng/mL), Dox +Cin (65±0.71ng/mL), EA (75.5±3ng/mL), and Dox+EA (64±1.4 ng/mL) treated groups were significantly lower than Dox-treated HCC group (122.3 ±3ng/mL).

**Alpha-L-fucosidase (AFU)**, has been anticipated as a promising tumor marker in the diagnosis of HCC. The administration of DEN/2-AAF produced a significant increase in serum AFU level (58.6 ± 0.27 U/L) when compared to normal rats (43.1± 0.41U/L). The level of AFU was decreased to (47.4± 0.66 U/L), (48.9± 0.73 U/L), and (44.4± 1.6) in rats treated with DOX, 1,8 cineole, and ellagic acid, respectively. Combination treatments of DOX+CIN (42.9± 0.32 U/L) and DOX+EA (41.7 ± 0.83U/L) showed a lower level of serum AFU compared to that of DOX alone (Fig 1C).

**Arginase-1 (Arg-1)**, a recent marker for HCC, is involved in the urea cycle leading to polyamines production and tumor cell proliferation. The serum level of arginase-1 in DEN/ 2-AAF-administered rats showed a significant increase to (358.6±8 U/L) as compared to normal rats (253.2 ± 5.3 U/L). Treated HCC rats with DOX, Cin, EA along with their combinations significantly decreased serum level of Arg-1 in a range from 286.9± 2.9 U/L to 299.5± 0.71 U/L as compared to the HCC group. Treatment combinations did not show any statistically different reduction in arginase-1 level as compared to the Dox group (Fig 1D).

## 1,8 cineole and ellagic acid reduced MiR-122 expression

MiR-122, a liver-specific miRNA, is a sensitive and specific marker of HCC. Therefore, the expression level of miR-122 in liver tissue was measured by qRT-PCR analysis (Fig 2) after the DEN/2-AAF induction of HCC. The expression of miR-122 was significantly down-regulated in the HCC group (0.2 ± 0.03) compared to the control group (1.01± 0.01). Treatment groups with Cin (0.72 ±0.03), EA (0.66 ± 0.08), and their combinations (0.58± 0.05 and 0.58 ± 0.01), respectively with Dox were resulted in dramatic upregulation of its expression as compared to the HCC group. However, combination treatments did not show any advantage in its expression compared to the Dox group (0.72 ± 0.06).

## Effect on EMT and its regulators

The levels of EMT key marker, vimentin, and EMT regulators, including TGF-β1, FSCN1, MMP-9, and VEGF were measured in liver tissue after DEN/2-AAF administration. Results from the western blot analysis indicated that DEN/2-AAF induced about 6.5, 4.8, 6- and 7.1-fold increases in vimentin, TGF-β1, FSCN1, and MMP-9 protein expression, respectively (Fig 3A). Similar behavior was observed at their levels in all treatment groups showing significant downregulation as compared to the HCC group. Nevertheless, combination treatments did not show any improvement as compared to the Dox group. Comparable results were obtained after determining the level of another EMT inducer, VEGF using ELISA kits. DEN/ 2-AAF induced a 2.5-fold increase in the VEGF level (761± 5.57 pg/mL) compared to the normal group (299 ± 6.24 pg/mL). All treated groups showed a significant reduction in the VEGF level compared to DEN/2-AAF administered group. However, there was no significant difference in its level between combination groups as compared to Dox treated group (284.67 ± 6.11 pg/mL) (Fig 3B).

## Histopathological findings

The liver of control rats showed the hepatic lobules with the typical hexagonal architecture together with a normal central vein, radiating hepatocytes showing regular cellular and nuclear size (Fig 4A). Normal rats showed the ordinary hexagonal architecture of the hepatic lobules with the normal portal tract that includes portal artery, portal vein, and bile duct (Fig 4B).

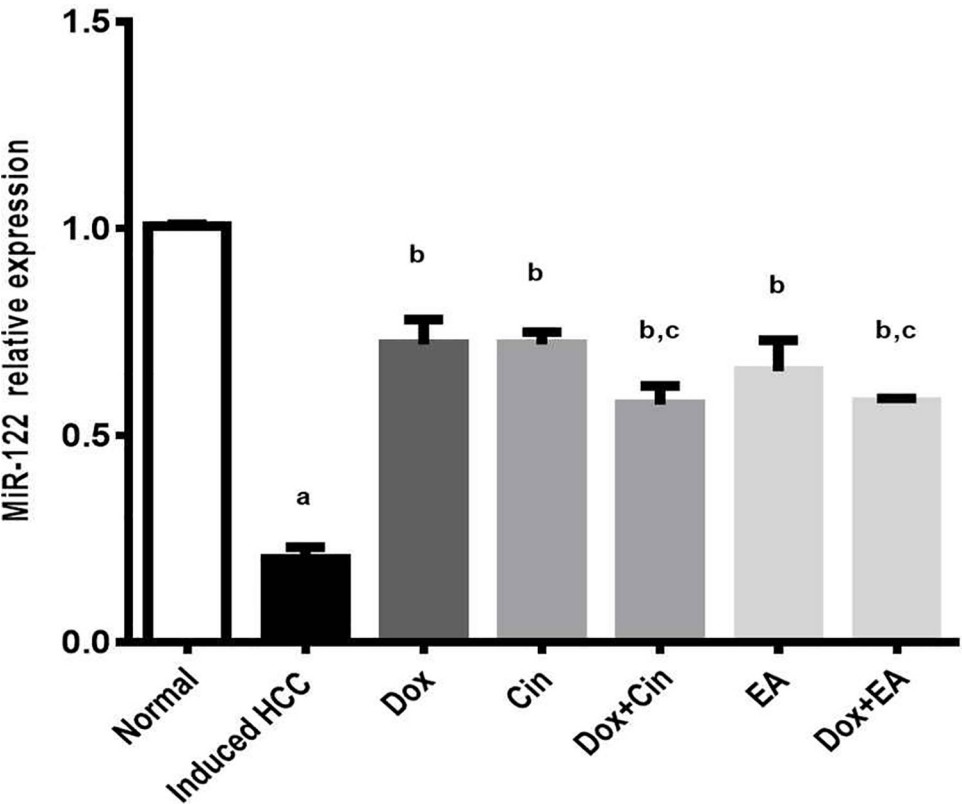

**Fig 2. MiR-122 expression using the qRT-PCR technique of DEN/2-AAF-induced HCC rats.** Rats were treated with 1,8 cineole (Cin), ellagic acid (EA), doxorubicin (DOX), and combinations. Data are presented as mean ± S.E.M, n = 10. [a] P<0.05 vs. normal group, [b] P<0.05 vs. DEN/2-AAF-induced HCC group and [c] P<0.05 vs. DOX treated group using one-way ANOVA followed by Tukey's post-hoc test.

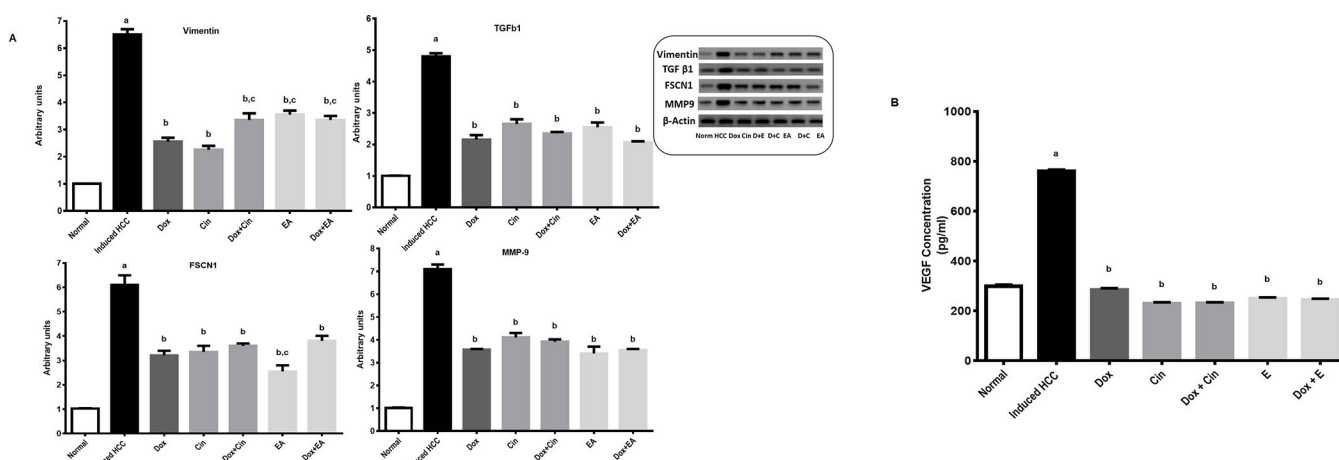

**Fig 3. EMT and its regulators in DEN/2-AAF-induced HCC rats.** (A) Representative Western Blot of Vimentin, TGF-β1, FSCN1, and MMP-9 after treatment with 1,8 cineole (Cin), ellagic acid (EA), doxorubicin (DOX), and combinations. Chemiluminescence analysis; optical densities were normalized to β-actin levels and expressed in arbitrary units. The data shown are representative of three independent experiments with comparable results. (B) VEGF concentration in liver homogenate using ELISA kits. Data are presented as mean ± S.E.M, n = 10. [a] P<0.05 vs. normal group, [b] P<0.05 vs. DEN/2-AAF-induced HCC group, and [c] P<0.05 vs. DOX treated group using one-way ANOVA followed by Tukey's post-hoc test.

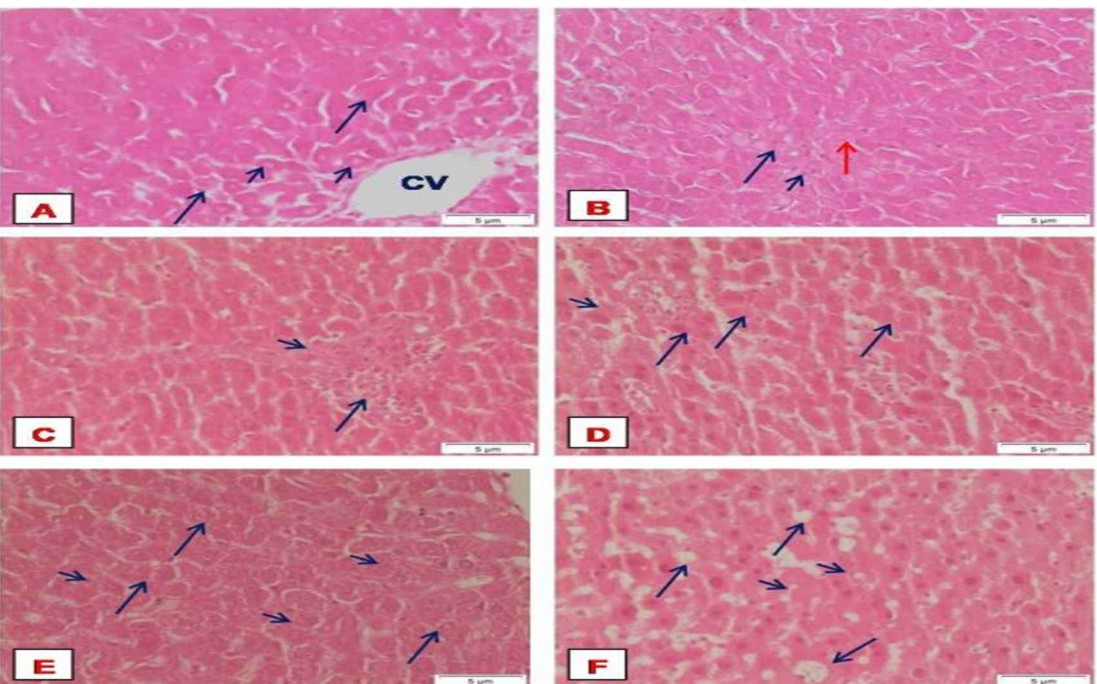

**Fig 4. Photographs of liver sections stained with H & E.** Normal rats showing (A) normal architecture with the central vein, normal hepatocytes (arrows) with normal nuclear size (arrowheads); (B) normal portal tract that includes portal artery (arrow), portal vein (arrowhead) and bile duct (red arrow). DEN/2-AAF induced rats showing (C) congestion of portal area (arrowhead) with an aggregation of inflammatory cells (arrow), (D) multinucleated hepatocytes (arrows), loss of radiating hepatocytes (arrowhead), (E) enlargement of hepatocytes (arrows), and nuclear size (arrowheads) (F) micro and macro-vesicular steatosis.

Microscopic examination of DEN+2-AAF administered rats showed aggregation of inflammatory cells in the portal areas (Fig 4C), multinucleated hepatocytes, loss of radiating hepatocytes (Fig 4D), enlargement of hepatocytes, and nuclear size as compared with normal control (Fig 4E) and micro/macro-vesicular steatosis (Fig 4F). The Dox-treated rats showed foci of enlarged cells were seen (Fig 5A), and the portal areas exhibited mild inflammatory infiltration (Fig 5B). On the other hand, rats treated with Cin or EA showed the ordinary structure of the hepatic lobules and portal areas (Figs 5C, 5D, 6C and 6D), respectively. The combined treatments of Cin or EA with Dox showed the ordinary structure of the hepatic lobule (Figs 5E and 6A), respectively. However, mild inflammatory infiltrations were observed in the portal areas of combined Cin treatment (Fig 5F), and congested portal areas associated with moderate inflammatory infiltration were observed in the portal areas of EA combined treatment (Fig 6B).

## Discussion

Hepatocellular carcinoma (HCC) is a very pervasive malignant disorder with multifaceted molecular pathogenesis. Studies proved that hepatocarcinogenesis had been linked with variations in several important cellular responses. Targeting molecules during the multistep process of hepatocarcinogenesis was of interest from the therapeutic perception because this may aid in a coup, delay, or even prevent tumorigenesis [29, 30]. Phytochemicals were shown to control HCC via interference with all phases of carcinogenesis, including initiation, promotion, and progression [31, 32]. Therefore, plant-derived compounds attract a lot of consideration because of their definite therapeutic effectiveness in the treatment of cancers [33, 34].

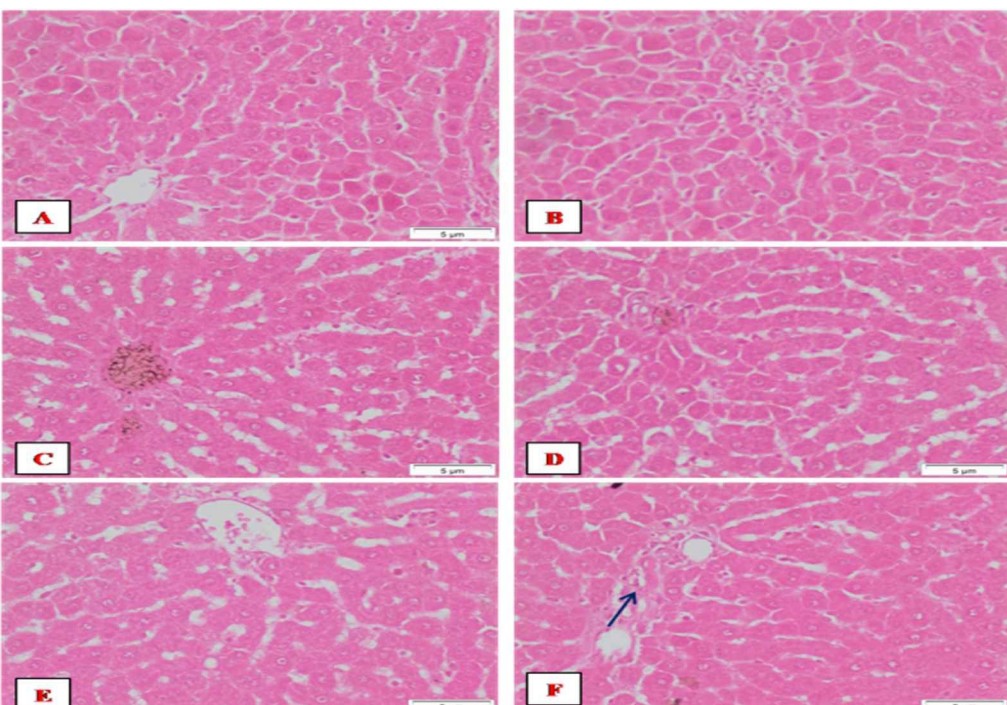

**Fig 5. Photographs of liver sections stained with H & E.** (A and B) Doxorubicin-treated rats showing a focal of enlarged cells with mild inflammatory infiltration, respectively. (C and D) 1,8 Cineole-treated rats were showing the ordinary structure of the hepatic lobule and portal area, respectively. (E and F) Dox+Cin treated rats showing the ordinary structure of the hepatic lobule with mild inflammatory infiltration, respectively.

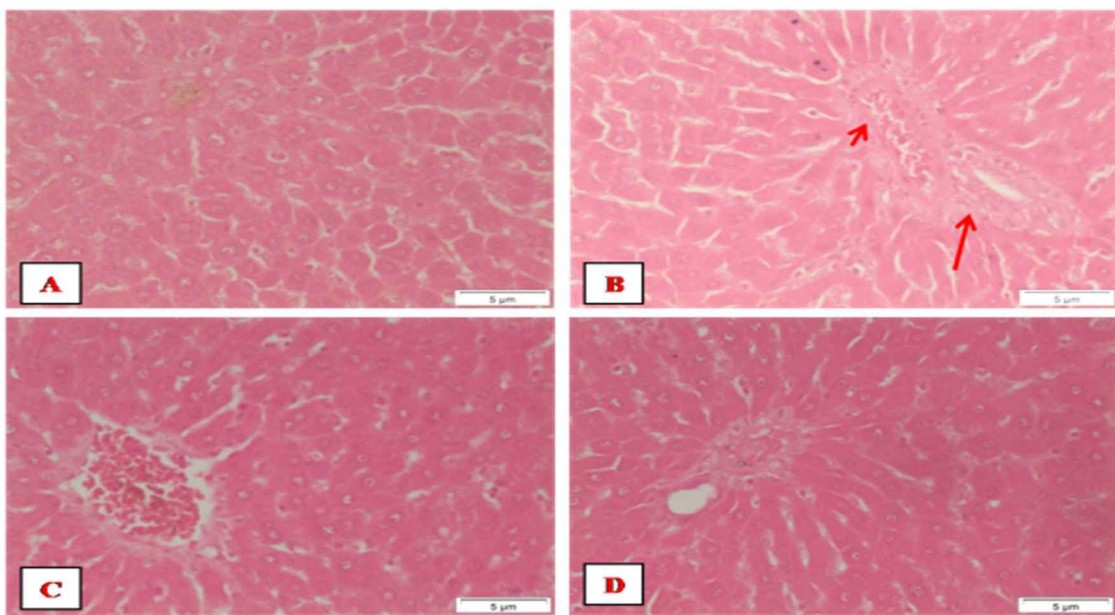

**Fig 6. Photographs of liver sections stained with H & E.** (A and B) Dox+EA treated rats showing the ordinary structure of the hepatic lobule and congested portal area (arrows) that were associated with moderate inflammatory infiltration (arrowheads). (C and D) Ellagic acid-treated rats showing the ordinary structure of the hepatic lobule and portal area, respectively.

Accordingly, the current work demonstrated for the first time the multi-target effects of two phytochemicals, 1,8 cineole (Cin) and ellagic acid (EA), against DEN/2-AAF-induced rat hepatocarcinogenesis.

DEN is the most used carcinogen to investigate hepatocarcinogenesis in rodents [35–37]. In models of experimentally induced HCC, initiation and promotion are essential steps. The initiation with DEN followed by using 2-AAF as a promoter was reported to produce successful development of experimentally induced HCC model [38, 39]. In the present work, DEN/2-AAF administration resulted in liver damage that was showed by an elevated level of liver function enzymes (ALT, AST, and ALP) along with impairment in the protein biosynthesis capacity. These abnormalities were observed remarkably in hepatoma and could be attributed to the hepatic lesions produced by DEN [35, 38, 40]. Treatment with 1,8 cineole and ellagic acid protected the liver from the effects of DEN plus 2-AAF on the previously mentioned markers of liver damage. These results are in agreement with an earlier study by our group showing the hepatoprotective effects of 1,8 cineole and ellagic acid [41] and demonstrated their ability to protect the liver from carcinogens by maintaining the cell membrane and functional integrity, thereby mitigating the progression of carcinogenesis.

Alpha-fetoprotein (AFP), alpha-L-fucosidase (AFU), arginase-1(Arg-1) are promising diagnostic tumor markers of HCC, and their measurements increased the detection sensitivity and specificity for HCC [42–44]. In the current study, disease progression was further confirmed by the elevated levels of these tumor markers in DEN/2-AAF administered rats, as well as the existence of multinucleated cells and steatosis that are common in early HCC [45, 46]. The marked decrease in levels of AFP, AFU, and Arg-1 along with the improvement of histopathologic features that were observed after treatment with 1,8 cineole and ellagic acid strongly suggested their anticancer activity against HCC.

One of HCC manifestations is iron overload demonstrated by elevation of serum ferritin and correlated with elevated serum aminotransferases proposing that ferritin came from damaged cells and was released into the serum [47, 48]. In accordance with these studies, we showed a significant elevation of serum ferritin after DEN/2-AAF that was overturned by treatment providing evidence for 1,8 cineole and ellagic acid protection against disturbances of iron metabolism in HCC. In addition, circulating microRNAs (miRNAs) have been implicated in hepatocarcinogenesis [49]. More importantly, microRNA-122 (miR-122) is a liver-specific miRNA constituting 70% of the total hepatic miRNAs, acting as a tumor suppressor, and its downregulation has been reported and correlated with HCC progression [50, 51]. In this investigation, our data showed for the first time that DEN/2-AAF induced a marked decrease in miR-122 expression while the therapeutic administration with 1,8 cineole and EA showed a significant improvement signifying the potential use of 1,8 cineole and EA in liver cancer treatment as multi-target anticancer agents.

The development of HCC is associated with the activation of epithelial-mesenchymal transition (EMT) that has been reported from both clinical and experimental observations [52]. Vimentin over-expression, a molecular characteristic of EMT, has been reported in cancers and correlated with increased tumor growth, invasion, and poor prognosis. Consequently, studies provided vimentin as an available targeted cancer therapy [53]. The decreased expression of vimentin in this study indicated for the first time that 1,8 cineole and ellagic acid might have a suppressive role on EMT in hepatocarcinogenesis. This finding comes to an agreement with previous reports displaying the anti-EMT effect of EA that was observed in breast cancer [54] and pancreatic cancer [33].

The activation of transforming growth factor-beta 1 (TGF- β1) and subsequent triggering of EMT is also critical in the development of HCC. It has been observed that TGF-β1-enriched cells showed an increase in the vimentin expression level [55]. There is also evidence

suggesting that FSCN1, an actin-binding protein involved in the invasion and migration, is overexpressed in response to TGF-β1 activation in HCC. Moreover, studies have demonstrated that FSCN1 also promoted EMT in cancers including HCC, and as predictable its suppression significantly suppressed vimentin expression, a key marker of EMT [10]. DEN was reported to induce hepatocarcinogenesis in rats via upregulating TGF-β1 signaling pathway [56]. In our study, we found a pronounced expression of TGF-β1, FSCN1, and vimentin in DEN/2AFF treated rats. These data suggested that TGF-β1 and, for the first time FSCN1 and vimentin are essential for the DEN/2-AFF induced hepatocarcinogenesis. Interestingly, our work showed primarily that 1,8 cineole and ellagic acid suppressed the expression of TGF-β1, FSCN1, and vimentin in this model providing new targets for both phytochemicals in HCC.

Elevated levels of matrix metalloproteinases (MMPs), especially MMP-9, are considered as an important factor in hepatocarcinogenesis, being a promoter of tumor invasion and angiogenesis as well [12, 57]. MMP-9 is overexpressed during EMT because of TGF-β1 activation [7]. Moreover, studies have demonstrated the functional interplay between MMP-9 and vascular endothelial growth factor (VEGF), another EMT inducer, in HCC, favoring tumor angiogenesis [12]. This can somewhat be explained by studies that demonstrated a positive correlation of MMP-9 and VEGF expression with the progression and recurrence of HCC [57, 58]. Angiogenesis has an important role in HCC progression and aggressiveness, being a part of its multifaceted molecular pathogenesis [13]. The potent suppressed expression of MMP-9 and VEGF was first observed in this study after 1,8 cineole treatment providing these two parameters as important targets of 1,8 cineole in combating HCC. Ellagic acid also decreased MMP-9 and VEGF expression. These findings were in line with earlier studies demonstrating the suppressive activity of EA on VEGF and MMP-9 [28, 59]. Convincingly, these results added potential for the role of 1,8 cineole and ellagic acid in chemoprevention and treatment of hepatocarcinogenesis.

Surprisingly, the different observed effects of 1,8 cineole and ellagic acid in the whole study were comparable to the effect of the potent chemotherapeutic agent, Doxorubicin, which is hindered by toxicity to normal tissues and tumor resistance [60]. However, the combination of both phytochemicals with Dox showed no additive effect regarding the investigated parameters.

## Conclusions

In conclusion, the anticancer effects of 1,8-cineole and ellagic acid were demonstrated for the first time through 1) Conserving of liver functions 2) Lessening of HCC tumor markers; alpha-fetoprotein, ferritin, arginase-1, and alpha-L-fucosidase. 3) Improvement in histopathologic features induced by DEN/2-AAF. 4) Upregulation of tumor suppressor microRNA; miR-122. 5) Declining of hepatocarcinogenesis mediators including EMT (vimentin) and EMT regulators (TGF-β1, FSCN1, MMP-9, and VEGF). These results highlighted the multiple-target effects of 1,8 cineole and ellagic acid in the treatment of HCC as potential therapeutic candidatures.

## Supporting information

**S1 Graphical abstract.**
(TIF)

**S1 Raw images.**
(ZIP)

## Author Contributions

**Conceptualization:** Heba M. I. Abdallah.

**Data curation:** Heba M. I. Abdallah, Sally A. El Awdan, Rehab F. Abdel-Rahman, Rasha M. Allam.

**Formal analysis:** Heba M. I. Abdallah, Sally A. El Awdan, Rehab F. Abdel-Rahman, Abdel Razik H. Farrag, Rasha M. Allam.

**Funding acquisition:** Heba M. I. Abdallah.

**Methodology:** Heba M. I. Abdallah, Sally A. El Awdan, Rehab F. Abdel-Rahman, Abdel Razik H. Farrag, Rasha M. Allam.

**Project administration:** Heba M. I. Abdallah.

**Software:** Heba M. I. Abdallah, Rasha M. Allam.

**Supervision:** Heba M. I. Abdallah, Sally A. El Awdan, Rehab F. Abdel-Rahman.

**Visualization:** Rehab F. Abdel-Rahman.

**Writing – original draft:** Rasha M. Allam.

**Writing – review & editing:** Heba M. I. Abdallah, Sally A. El Awdan, Rehab F. Abdel-Rahman, Abdel Razik H. Farrag.

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
