## [Editor Report · Decision Letter 0]

20 Jul 2021

PONE-D-21-18911

1,8 cineole and ellagic acid inhibit hepatocarcinogenesis via upregulation of

MiR-122 and suppression of TGF-β1, FSCN1, vimentin , VEGF, and MMP-9

PLOS ONE

Dear Dr. Rasha M Alam,

Thank you for submitting your manuscript to PLOS ONE. After careful consideration, we feel that it has merit but does not fully meet PLOS ONE’s publication criteria as it currently stands. Therefore, we invite you to submit a revised version of the manuscript that addresses the points raised during the review process.

We look forward to receiving your revised manuscript.

Kind regards,

Vikas Kumar, Ph.D

Academic Editor

PLOS ONE

Additional Editor Comments:

Well performed and well written manuscript. But language is the major issue and i suggest to send the manuscript for language editing company for correction.

Journal Requirements:

3. In your Methods section, please provide additional information on the animal research and ensure you have included details on : (1) how often the animal health was monitored (2) whether any animals died during induction of hepatocarcinogenesis, and if so, how many were found dead? (3) specific criteria of how animal health was monitored.
---

## [Author Response · Author response to Decision Letter 0]

16 Aug 2021

Additional Editor Comments:

• Well performed and well written manuscript. But language is the major issue and i suggest to send the manuscript for language editing company for correction.

Response: We appreciated the editor's comment. Accordingly, we sent the manuscript to a language editing company and the proof was attached.

Journal Requirements:

Response: We respect the journal requirements, and the corrections have been done.

Response: We appreciated the reviewer's comment. We revised the whole references list and added four references to be a final of 60. All the changes have been done to meet the journal requirement.

The added references are listed below:

1. Ferlay J, Colombet M, Soerjomataram I, Mathers C, Parkin DM, Pineros M, et al. Estimating the global cancer incidence and mortality in 2018: GLOBOCAN sources and methods. Int J Cancer. 2019;144(8):1941-53. Epub 2018/10/24. doi: 10.1002/ijc.31937. PubMed PMID: 30350310

2. Sobreira Dantas Nobrega de Figueiredo FR, Monteiro AB, Alencar de Menezes IR, Sales VDS, Peticia do Nascimento E, Kelly de Souza Rodrigues C, et al. Effects of the Hyptis martiusii Benth. leaf essential oil and 1,8-cineole (eucalyptol) on the central nervous system of mice. Food Chem Toxicol. 2019;133:110802. Epub 2019/09/08. doi: 10.1016/j.fct.2019.110802. PubMed PMID: 31493462.

3. Abdel-Hamid NM, Abass SA, Mohamed AA, Muneam Hamid D. Herbal management of hepatocellular carcinoma through cutting the pathways of the common risk factors. Biomed Pharmacother. 2018;107:1246-58. Epub 2018/09/28. doi: 10.1016/j.biopha.2018.08.104. PubMed PMID: 30257339

4. Choudhari AS, Mandave PC, Deshpande M, Ranjekar P, Prakash O. Phytochemicals in Cancer Treatment: From Preclinical Studies to Clinical Practice. Front Pharmacol. 2019;10:1614. Epub 2020/03/03. doi: 10.3389/fphar.2019.01614. PubMed PMID: 32116665

3. In your Methods section, please provide additional information on the animal research and ensure you have included details on : (1) how often the animal health was monitored (2) whether any animals died during induction of hepatocarcinogenesis, and if so, how many were found dead? (3) specific criteria of how animal health was monitored.

Response: We are thankful for the reviewer's accuracy and in order to respond to the reviewer comment, we added the specific criteria of animal health and the number of dead animals during induction of hepatocarcinogenesis in the method section.

(1) The general health conditions of animals concerning food consumption, water intake, and body weight were observed during all the experimental time besides signs of distress including respiratory changes, body temperature, or even unusual behavior were also observed. We did not notice a significant variation in the previous parameters between the experimental groups except for the DEN/2-AAF group that showed a modest weight loss compared with the others (Page 5, lines from 100 to 105).

(2) Throughout the initiation phase of DEN treatment (first eight weeks), only one rat was dead. After starting 2-AAF administration (promotion phase), another two rats from the DEN/2-AAF induced HCC group died (Page 5, lines 111-113).

Response: We understand the journal requirement and accordingly we uploaded a separate PDF file with the raw data of the original uncropped blot images. 

Response: We respect the reviewer's comment. So, the ethics statement is only now in the method section.

---

## [Editor Report · Decision Letter 1]

11 Oct 2021

1,8 cineole and ellagic acid inhibit hepatocarcinogenesis via upregulation of

MiR-122 and suppression of TGF-β1, FSCN1, vimentin , VEGF, and MMP-9

PONE-D-21-18911R1

Dear Dr. Rashna M. Allam

We’re pleased to inform you that your manuscript has been judged scientifically suitable for publication and will be formally accepted for publication once it meets all outstanding technical requirements.

Kind regards,

Vikas Kumar, Ph.D

Academic Editor

PLOS ONE

Additional Editor Comments (optional):

All suggestion included in the manuscript.
---

## [Editor Report · Acceptance letter]

28 Oct 2021

PONE-D-21-18911R1 

1,8 Cineole and Ellagic acid inhibit hepatocarcinogenesis via upregulation of
MiR-122 and suppression of TGF-β1, FSCN1, Vimentin, VEGF, and MMP-9 

Dear Dr. Allam:

I'm pleased to inform you that your manuscript has been deemed suitable for publication in PLOS ONE. Congratulations! Your manuscript is now with our production department. 

Kind regards, 

on behalf of

Dr. Vikas Kumar 

Academic Editor

PLOS ONE